# R-PCR: Recurrent Point Cloud Registration Using High-Order Markov Decision

Xiaoya Cheng [†], Shen Yan [†], Yan Liu, Maojun Zhang and Chen Chen *

College of Systems Engineering, National University of Defense Technology, Changsha 410000, China
* Correspondence: chenchen16@nudt.edu.cn
† These authors contributed equally to this work.

**Abstract:** Despite the fact that point cloud registration under noisy conditions has recently begun to be tackled by several non-correspondence algorithms, they neither struggle to fuse the global features nor abandon early state estimation during the iterative alignment. To solve the problem, we propose a novel method named R-PCR (recurrent point cloud registration). R-PCR employs a lightweight cross-concatenation module and large receptive network to improve global feature performance. More importantly, it treats the point registration procedure as a high-order Markov decision process and introduces a recurrent neural network for end-to-end optimization. The experiments on indoor and outdoor benchmarks show that R-PCR outperforms state-of-the-art counterparts. The mean average error of rotation and translation of the aligned point cloud pairs are, respectively, reduced by 75% and 66% on the indoor benchmark (ScanObjectNN), and simultaneously by 50% and 37.5% on the outdoor benchmark (AirLoc).

**Keywords:** large-scale point cloud; high-order markov decision; point cloud registration; recurrent neural network

## 1. Introduction

Point clouds are 3D representations of real-world objects or scenes, generated by capturing the coordinates of numerous points in the scene using laser scanners or depth cameras. The registration of these point clouds is a fundamental task in autonomous vehicles [1], augmented reality (AR) [2], and 3D reconstruction [3]. Point cloud registration becomes challenging when dealing with large datasets that contain millions or billions of points, which is common in real-world scenarios. The process involves finding the optimal transformation that aligns the overlapping regions of two or more point clouds while minimizing the registration error.

State-of-the-art approaches to point cloud registration commonly rely on finding 3D–3D local correspondences between point cloud pairs [4–9]. The resulting relative transformation is then calculated within a robust estimator, e.g., RANSAC (random sample consensus) [10]. Correspondence-based methods are capable of achieving high accuracy in registration. By taking multiple correspondences, the methods could identify and ignore erroneous or spurious matches, estimating transformation primarily via inliers. Although correspondence-based methods are generally robust to noise and outliers, challenges still remain when dealing with noisy, ambiguous data. This is due to the constraints of 3D acquisition devices, which induce errors caused by sensor noise and system error during subsystem integration. For example, LiDAR (light detection and ranging) sensors may have a limited resolution or sensitivity due to electrical or electronic noise in the system. Simultaneously, finding correspondences in large or complex datasets is computationally expensive and time-consuming. Even though the soft correspondence [11] employs a weighted approach that considers the similarity between points and their surrounding neighborhoods, it is still challenging to measure similarity and get point–point correspondences when facing the point clouds that contain noise, outliers, and density differences.

Recently, several deep learning methods have been developed to extract a global feature embedding from point cloud pairs and facilitate registration by aligning these features, namely non-correspondence registration. Non-correspondence registration is a class of methods used to align or register two or more point clouds without explicitly establishing point correspondences between them. Examples including global feature approaches [12–16] are proposed, which exhibit an excellent potential for point registration, especially under noisy conditions. Instead of extracting local features and building explicit correspondences, such methods directly determine relative transformation between pairwise point clouds using a neural network. The feature descriptor captures the shape, distribution, or other attributes. Unlike those that rely on point-to-point correspondences, non-correspondence registration aims to establish the global alignment of point clouds by leveraging the inherent structure and global features, which is particularly effective when point correspondences are difficult to match due to occlusions or noise in the point clouds.

A limitation of the non-correspondence registration is that the feature extraction networks nonlinearly increase the input data to high-dimensional global feature space and maintain insufficient robustness with such implicit correspondence information. The lack of interpretability of these networks makes it challenging to understand the relationships between the extracted features and the underlying geometric properties of the point clouds. This can limit the performance of the registration algorithm and hinder the ability to generalize to new datasets or scenarios. Therefore, the optimization is considered to use an iterative algorithm to improve robustness.

Recent works [11,13,14,17,18] decomposes the final transformation estimation into a sequence of iterative update steps, which can refine the initial estimate of transformation parameters through multiple iterations, leading to better alignment accuracy. However, the existing methods come with their own drawbacks. First, most of the previous works abstract the relative transformation via global embedding and concatenation, with no feature fusion between point cloud pairs. Second, the iterative process is commonly regarded as a first-order Markov decision process (MDP), where an update transformation is entirely determined by the current state of point cloud pairs without knowledge of the previous state. Unfortunately, such iterative adjustment disregards the sequential property of the transformation update. The registration step is uncertain and controversial during iterative refinement, but we find there is a relative state constraint that emerges the overall registration of view. That is, a robust registration system has strong stability, which could ensure a similar movement trend towards the target state for each minor adjustment. Since first-order MDPs only consider the current state and action when computing the movement probabilities, they may have limitations in capturing long-term dependencies among substeps. This would lead to suboptimal policies that do not take into account the feature consequences of current decisions.

In this paper, we introduce a novel deep network architecture, R-PCR (recurrent point cloud registration), that can not only effectively fuse independent global features, but also integrate the high-order Markov decision into iterative point registration. Specifically, we first apply PointNet [19] structured network as an embedding function to separately extract global geometry information for the source and target point cloud. Then, to improve the expression ability of global features and introduce possible implicit correspondence, we propose a lightweight cross-concatenation module and a large-receptive network to merge information between pairwise point clouds. Finally, a recurrent GRU-based (gate recurrent unit) [20] update operator is used to bring a high-order state from the previous movement, iteratively updating the estimated transformation. The interrelated constraints between substeps are use to model the high-dimensional state and action spaces. By adding such constraints, our approach can be more expressive and better able to model complex registration tasks, particularly noise-afflicted data. This allows for more accurate registration, which is critical for ensuring the stable convergence.

We evaluate the proposed method on several standard point cloud registration datasets, including synthetic data (ModelNet40) and real data (ScanObjectNN). Furthermore, we

quantitatively show the effectiveness of our approach on large urban data (AirLoc). The experiments show that R-PCR outperforms global-based registration baselines by a large margin.

To summarize, our main contributions are threefold:

- We introduce R-PCR, a novel deep network architecture for point cloud registration that effectively fuses independent global features and integrates high-order Markov decision into iterative point registration.
- We propose a simple yet effective cross-concatenation module and large-receptive network to enhance the feature fusion between pairwise point clouds, improving the expression ability of global features. This allows more accurate registration, particularly for noise-afflicted data, and ensures stable convergence.
- R-PCR shows superior performance on several standard point cloud registration datasets, including synthetic and real data, as well as large urban data. Our method outperforms global-based registration baselines by a large margin.

## 2. Related Work

### 2.1. Point Cloud Registration

Traditional methods [21–25] treat point registration as an energy minimization problem which usually defines a distance function and provides a closed-form solution. For example, given source and target point clouds, the classic algorithm ICP( iterative closest point) [21] is an iterative process that starts with an initial estimate of the transformation between the point clouds and then refines this estimate in each iteration until convergence. The basic idea of ICP is to find correspondences between the points in the two point clouds and use these correspondences to estimate the transformation parameters that align the point clouds. Apart from the point–point fashion, there are also many ICP-variant methods, such as the point–line [22] and point–plane [26] methods. Although the ICP algorithm is a popular method for point cloud registration for the simplicity of implementation, it is sensitive to initial conditions and becomes stuck in local minima.

Recently, learning-based methods have been widely used [4,27,28]. Compared with classical ones, they express better feature representation and faster computing ability. Ref. [27] uses smoothed densities to estimate the probability of each point being a match for a given point in the other point cloud, and iteratively finds the optimal transformation between the point clouds that maximizes the probability of the corresponding point pairs. Ref. [28] proposes a new local feature descriptor of point clouds, namely the point pair feature (PPF) which captures the geometric relationship between a point and its neighboring points in the global context of the scene. Ref. [6] attempts to use a probabilistic embedding of local features to estimate the transformation between the point clouds, and shows the effectiveness in terms of point clouds with low overlap. Ref. [4] applies a novel feature aggregation strategy to produce a compact and discriminative feature descriptor for each point, which jointly learns to detect and describe local features using a dense 3D convolutional network. Although they have endured a significant development in correspondence-based registration, most of the existing methods rely on local feature descriptors for geometric representation, which leads to limited robustness for sensitive to noise and outliers, especially for complex or large-scale scenes.

As a result, another line of research explores whether we could address point cloud registration through a non-correspondence pipeline [12–14,18]. Much research [12,13,18] have been carried out on global features since the first empirical research of PointNet [19]. Ref. [13] initiatively learns a global feature representation via PointNet and extends the classical Lucas–Kanade algorithm [29] to operate directly on point clouds. Ref. [12] shares the belief that classical alignment techniques for aligning the PointNet features are not robust enough to noise and proposes a fully differentiable architecture to refine the registration estimate. Ref. [30] attempts to extract distinctive features and uses a feature-metric function to compare and align the point cloud pairs under semi-supervision. Ref. [31] leverages recent advances in implicit neural representations to learn a continuous

and equivariant function that maps points from one point cloud to the other, which could improve the registration robustness against noise in point clouds with the help of implicit shape learning. Ref. [18] uses imitation learning to learn from expert demonstrations and then fine-tunes the learned policy using reinforcement learning. The imitation learning phase involves training a neural network to predict transformation in discrete step sizes. All these methods adopt a Siamese network as a feature extraction tool, and find differences in the high-dimensional feature space for relative transformation regression. One significant drawback of these methods is that they use a separated training process to learn a stand-alone feature extraction network. Differently from this kind of method, we attempt to extract distinctive features and use a cross-concatenation module to fuse the feature space and compare point clouds in a receptive field.

### 2.2. Iterative Refinement

Iterative refinement is a process used in machine learning algorithms to increase the accuracy of the results. It involves repeatedly tweaking the model parameters in order to find the parameters that will reduce the error rate in the model. Iterative refinement has been used in many tasks such as image matching [29], object recognition [32], and so on. In each of these tasks, the initial estimate may be obtained using a rough or approximate method, and the iterative refinement process is used to refine the estimate until a high level of accuracy is achieved. Iterative refinement has several advantages over other methods, such as its ability to handle large amounts of data, the ease of implementation, and the ability to handle non-linear optimization problems.

To find an optimal transformation, current works commonly follow the iterative idea [11–13,18], which decomposes the rigid transformation multiple smaller easy-achieving steps. Iterative refinement is especially useful in combination with high-order Markov models, as using multiple orders allows for the model to make more accurate predictions. However, they all define their registration framework as a first-order Markov process, where the future state of the object only depends on its current state and is independent of its history. In contrast, our approach treats registration process as a temporal correlation task and we construct a high-order Markov model to introduce earlier state information. We solve the problem via a recurrent GRU-based [20] update operator. Instead of using a simple linear transformation, which only considers the spatial relationship between current point cloud state, a high-order Markov model is used to model complex registration tasks. For large-scale scenes, this allows for more stable convergence, robust to noise and outliers.

### 2.3. Recurrent Neural Networks

Recurrent neural networks (RNNs) are a type of artificial neural network used in machine learning applications that exhibit temporal behavior; that is, they process data in sequence or over time. By including feedback connections and memory they are able to store and use temporal information received at different points in time, allowing them to recognize patterns in long-term data sequences.

Recurrent neural networks (RNNs) have been shown to be very powerful in many applications such as natural language processing [33,34], video prediction [35,36], and other domains [37–39]. RNNs enable computing within an ongoing process, making them inherently suitable for capturing temporal dynamics in vision tasks such as activity recognition, saliency prediction, and attribute prediction. They could also learn long-term dependencies explicitly and identify temporal correlations, which are particularly important for recognizing temporal patterns in temporal data. Moreover, due to their ability to leverage contextual information from multiple frames, RNNs are well suited for solving problems that involve spatiotemporal reasoning such as motion tracking [40] and optical flow estimation [39]. RAFTs (recurrent all-pairs field transforms) [39] outline an end-to-end approach to motion tracking and optical flow estimation using recurrent neural networks. It proposes a model based on an iterative field transform that combines multiple-input layers with intermediate connections.

Recurrent models [20,41], especially with gated units such as GRU (gate recurrent unit) [20], are widely applied in modeling temporal sequences. GRU uses two different gates to control the information flow: the update gate, which determines how much of the previous memory cell's state to keep, and the reset gate, which determines how much of the new input information to incorporate into the memory cell. This makes them more efficient in learning how to interpret the input information. The spatiotemporal models with GRU units not only focus on the current statement but learn continuous variations in the past motion sequences, which effectively capture both long-term and short-term dependencies in data, and are useful for a variety of sequence-based tasks. To this end, we apply GRU units in the iterative registration pipeline, effectively integrating the current state and capturing long-term temporal dynamics.

## 3. Approach

The overview of the proposed method is exhibited in Figure 1. Given Source point $X$ and Target point $Y$, we aim to estimate the 6-DoF transformation. To achieve this goal, our method can be distilled down to three stages: (1) point cloud feature embedding (Section 3.2), (2) global feature fusion (Section 3.3), and (3) iterative updates (Section 3.4), where all stages are differentiable and composed into an end-to-end trainable architecture.

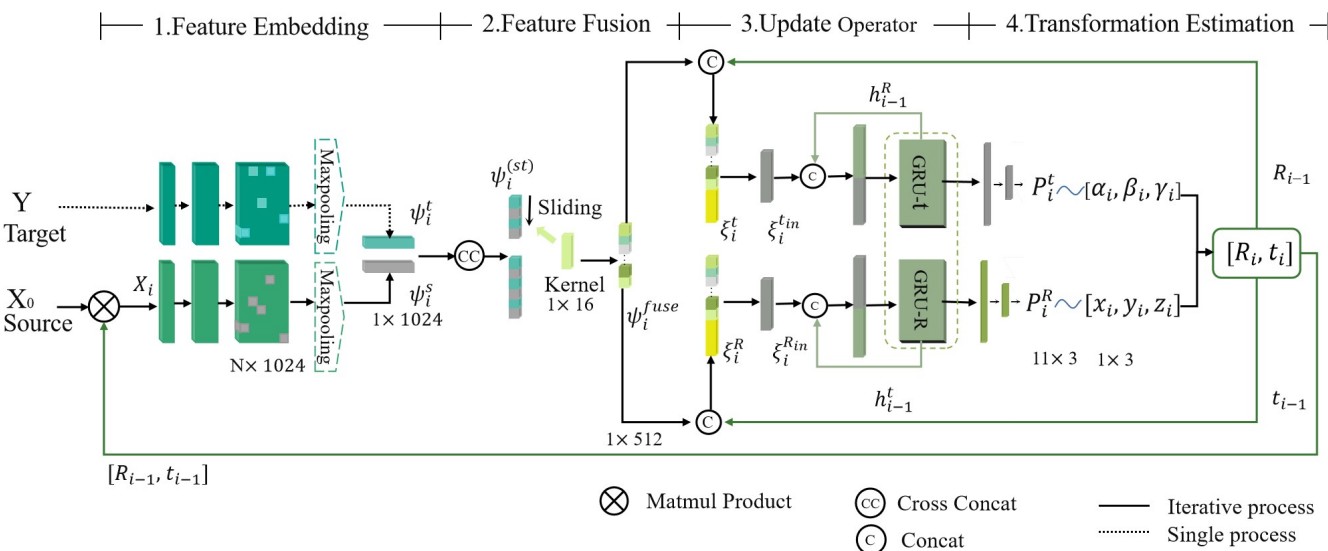

**Figure 1. Overview of the proposed method.** The PointNet-based network first separately extracts the global feature vector($\psi_i^s$, $\psi_i^t$) for the current source $X_i$ and the target $Y$. Then, the feature fusion block fuses global pair-wise features by cross concatenation and learns implicit correspondence ($\psi_i^{fuse}$) with a 16-sized kernel. After that, the update operator utilizes two GRU units to maintain long-term state information. The input feature is taken as the concatenation of $\psi_i^{fuse}$, the $(i-1)$th-obtained transformation, and hidden state $h_{i-1}$. Finally, the estimation network is trained to predict a probability distribution of transformation in a sequence using fully connected layers and return the current transformation $[R_i, t_i]$.

### 3.1. Preliminaries

In real scenes, the point cloud captured by the sensor is generally noise-afflicted due to various external signal interference. Therefore, we consider a pair of 3D point clouds, Source $X$ and Target $Y$, modeling the same scene, which shares an incomplete point-to-point correspondence. A rigid transformation $T^* = [R^* \in SO(3), t^* \in \mathbb{R}^3]$ exists, that aligns point sets,

$$Y = R^* X + t^*. \tag{1}$$

where $R^*$ is a rotation matrix and $t^*$ is a translation vector. As our method aims to compute the optimal transformation by multiple iterative steps. We use $X_{0:k} = (X_0, \ldots, X_k)$ to

denote the modified Source point from step 0 to $k$. During each step $i$, we estimate a rigid transformation $\Delta T_i = [\Delta R_i \in SO(3), \Delta t_i \in \mathbb{R}^3]$ between the Source $X_{i-1}$ and the Target $Y$.

Our approach is based on recurrent refinement. The optimization process is expressed as a sequence of iterative update steps. During each step i, we estimate a rigid transformation $\Delta T_i$ between the current source $X_i$ and the target $Y$, and then update the current transformation $T_i$ and source $X_i$,

$$T_i = \Delta T_i \times T_{i-1}, \tag{2}$$

$$X_i = \Delta T_i \times X_{i-1}. \tag{3}$$

Suppose we take n steps to compute the transformation between the initial source $X_0$ and the target $Y$, the final estimate after n steps is

$$X_n = \Delta T_n \times \ldots \Delta T_1 \times X_0. \tag{4}$$

### 3.2. Feature Embedding

Given two point clouds, the initial source $X_0$ and the target $Y$, we use a weight-sharing network to extract features from the two point clouds. We choose a PointNet-like architecture [19] to encode the point geometric shape. The feature embedding network $f_\theta$ is applied to both the source $X$ and the target $Y$, respectively. We only use 1D convolution layers of size [64, 128, 1024] without T-nets to transform the spatial position of raw point clouds into high feature space. To guarantee the disorder property of point clouds, we use max pooling to obtain a global feature vector, $\mathcal{R}^{N \times 3} \to \mathcal{R}^{1 \times M}$, where $N$ is point cloud samples, $M$ is output channel feature vector. After that, we obtain a pair of embedding feature vectors, source $\psi_i^{(s)}$ and target $\psi_i^{(t)}$.

### 3.3. Feature Fusion

For a certain step $i$, suppose we obtain the source feature $\psi_i^{(s)} = \{\psi_1^{(s)}, \ldots, \psi_M^{(s)}\}$ and the target $\psi_i^{(t)} = \{\psi_1^{(t)}, \ldots, \psi_M^{(t)}\}$, consider that the direct concatenation of these M dimensional global features brings too much weak global feature representation ability, which does not utilize the feature correspondence of two global features.We utilize the particularities of the Siamese network, that is, feature vectors $\psi_i^{(s)}$ and $\psi_i^{(t)}$ include a large number of feature pairs extracted by the same embedding function. Therefore, we further design a cross concatenation module to fuse the features into a $1 \times 2M$ feature $\psi_i^{(st)} = \{\psi_1^{(s)}, \psi_1^{(t)}, \ldots, \psi_M^{(s)}, \psi_M^{(t)}\}$ under the same network mapping relationship. In the following we use a 16-sized kernel with large receptive field to get a global fused feature depending on the particularities of X and Y together. The experiments show that the large and shallow convolution layer performs better than small and deep convolution layer in the process of local feature extraction; the detail can be seen in Section 4.6.

### 3.4. Recurrent Refinement

To simulate the high-order Markov decision process, we take advantage of a recurrent network to estimate a sequence of transformation $\{T_0, \ldots, T_k\}$, starting from $R_0 = I^{3 \times 3}$ and $t_0 = 0$. Each iteration produces an update direction $\Delta T$ which is applied to the current estimate: $T_i = \Delta T \cdot T_{i-1}$.

3.4.1. Disentangled Transformation

When the optimal transformation is computed by multi-step, it is regarded as iterative registration. During each step i, we estimate a rigid transformation $\Delta T_i = [\Delta R_i \in SO(3), \Delta t_i \in \mathbb{R}^3]$ between the source $X_{i-1}$ and the target $Y$, and then update the current transformation $T_i$ and source $X_i$. The standard representation of transformation is updated by:

$$R_i = \Delta R_i R_{i-1}, \; t_i = \Delta R_i t_{i-1} + \Delta t_i \tag{5}$$

$$X_i = R_i X_0 + t_i. \tag{6}$$

As Equation (5) show that the standard representation of transformations typically updates translation by two variables, making it difficult to modify the translation or rotation independently. We would like the rotation not to induce an additional the point cloud translation. In our experiments, we use a disentangled representation, the rotation and translation are separated into separate variables, which can be optimized individually. To disentangle rotation and translation, we use mathematical models that describe the transformation of an object from one coordinate system to another. Following the idea [18], the $T_i = [R_i, t_i]$ is given by:

$$R_i = \Delta R_i R_{i-1}, \ t_i = \Delta t_i + t_{i-1} \tag{7}$$

and the current source $X_i$ is updated by

$$X_i = R_i(X_0 - \mu_{X_0}) + \mu_{X_0} + t_i \tag{8}$$

where $\mu_{X_0}$ is the centroid of the initial source. During the transformation update, the origin of the coordinate system is offset to the centroid of the initial source $X_0$. In this way, the rotation will not change the translation of point cloud and the network could update the iterative transformations independently.

### 3.4.2. Update Operator

To effectively distinguish rotation and translation, we design two branch networks. The rotation branch returns Euler angles in radians, and the translation branch outputs the axis movement of the coordinate system. The network architecture is detailed in Figure 2.

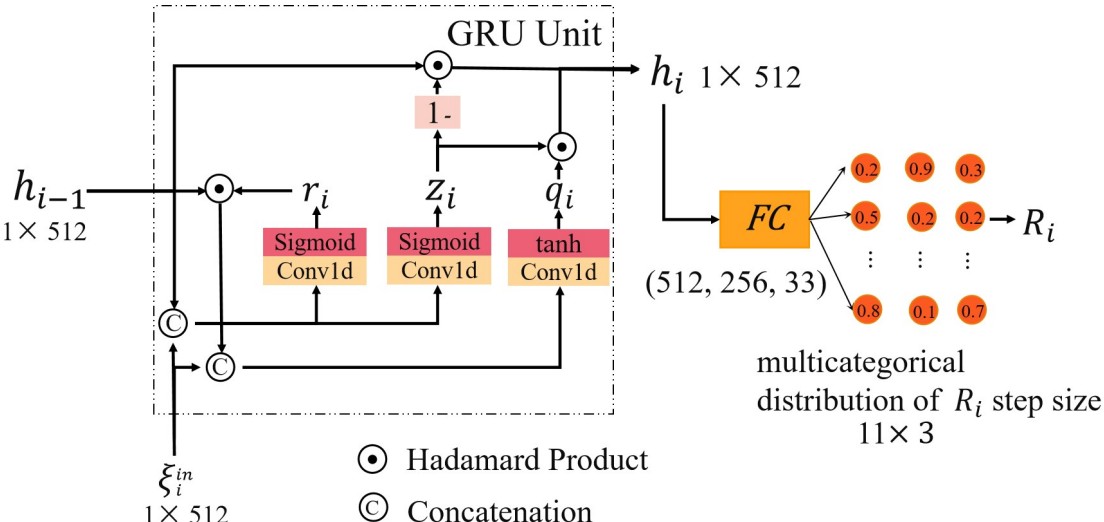

**Figure 2. Architecture of recurrent refinement.** A core component of a recurrent refinement module is a gated activation unit. The hidden state $h_{i-1}$ outputted by the GRU is passed through three fully connected layers, each with dimensions [512, 256, 33]. The output layer outputs the multicategorical distribution of the rotation and takes the largest possible step as estimated rotation $R_i$.

Given the fused feature $\psi_i^{fuse}$ at step $i$, the update operator takes transformation $(R_{i-1}, t_{i-1})$, state abstraction $\xi_i^{fuse}$, and a latent hidden state $h_{i-1}$ as input $\xi_i^{in}$, and outputs the pose update $(\Delta R_i, \Delta t_i)$ and an updated hidden state $h_i$. A core component of the update operator is a gated activation unit based on the GRU cell, whose internal structure is shown as follows:

$$z_i = \sigma(Conv1d([h_{i-1}, \xi_i^{in}], W_z)) \tag{9}$$

$$r_i = \sigma(Conv1d([h_{i-1}, \xi_i^{in}], W_r)) \qquad (10)$$

$$\tilde{h}_i = tanh(Conv1d([r_i \odot h_{i-1}, \xi_i^{in}], W_h)) \qquad (11)$$

$$h_i = (1 - z_i) \odot h_{i-1} + z_i \odot \tilde{h}_i \qquad (12)$$

The symbol $\odot$ represents the Hadamard product. The activation function of GRU augments the usual logistic sigmoid activation with two gating units called reset $r$ and update $z$ gates, which dynamically inject previous hidden information.

### 3.4.3. Transformation Estimation

The hidden state $h_i^R$ and $h_i^t$ outputted by the GRUs are passed through two fully connected layers to recover pose transformation $SE(3)$. In our experiments, taking an exact registration as output in each iteration may result in the divergence of the entire registration process, which allows us to adapt the action space to the current state of the registration process.

Therefore, we robustify the update transformation steps using discrete, limited step sizes. The rotation and translation are updated in a decoupling format, and each dimension is estimated separately. The decoupling estimation allows a stability of iterative registration process. For example, if we have already reached a good estimate in one dimension, we can reduce the number of steps along this dimension and increase it in other dimensions. For translation, step sizes are interpreted in meters. For rotation, step sizes are interpreted in angle. The action space includes 11 step sizes per axis in an exponential scale: [0.0033, 0.01, 0.03, 0.09, 0.27] in positive and negative directions, as well as a "stop" step. As a result, we invert the pose estimation into a multiple classification problem. As shown in Figure 2, the refinement module outputs a multicategorical distribution of each axis for rotation and translation. During training, the model is trained to predict possibility of step sizes and adopted cross-entropy loss for supervision. The model is optimized to select the update step that will lead to the registration in the most stable movement. We obtain the ground truth transformation at each step by the following disentangle relative pose computation:

$$\Delta R_i^* = R^* R_{i-1}^{-1} \qquad (13)$$

$$\Delta t_i^* = t^* - t_{i-1} \qquad (14)$$

## 4. Experiments

To demonstrate the effectiveness of the proposed approach, we conduct a qualitative and quantitative analyses on three point cloud datasets, including indoors and outdoors. Additionally, we perform ablation studies to determine the contribution of each component of the R-PCR to its overall effectiveness. Detailed comparisons and analysis of the results are presented.

### 4.1. Datasets

ModelNet40: ModelNet40 [42] is a large-scale dataset of 3D CAD models, consisting of 40 different object categories from each of 12,311 CAD models.The objects in ModelNet40 range from everyday household items such as chairs and desks to more complex objects such as airplanes and boats. Each object is represented as a point cloud, which is a set of 3D points that describe the surface geometry of the object. The point clouds are normalized, aligned, and sampled to have a fixed number of points, making them suitable for deep learning-based approaches.

ScanObjectNN: ScanObjectNN [43] is a comprehensive 3D object recognition and pose estimation dataset consisting of 15 object categories commonly found in indoor environments. It contains over 15,000 high-resolution 3D point clouds with detailed annotations of objects such as cars, pedestrians, and cyclists. Each object instance is labeled with its 3D pose obtained through laser scanning and a robotic arm.

AirLoc: AirLoc [44] is a large urban dataset built from the multi-sensor acquisition. We test our approach on the laser-scanned part, which is highly detailed 3D point cloud data gathered by Airborne LiDAR that represents the surface of an outdoor area in Changsha city. AirLoc was gathered by a DJI M300 (https://www.dji.com/cn/matrice-300, accessed on 23 Match 2023), carried DJI L1 (https://www.dji.com/cn/zenmuse-l1, accessed on 23 Match 2023) laser scanner and contained about 400M points, approximately 640,000 m$^2$. The data provides a highly detailed view of the terrain and any objects, containing different natural and artificial scenarios such as buildings, roads, plants, etc. We visualize the laser-scanned map in Figure 3. The whole laser-scanned map is used to evaluate our model, split into submaps at fixed intervals of 100 m and resulted in 60 test submaps.

### 4.2. Implement Detail

For a fair comparison between baselines and our model, all approaches are trained on the training split of ModelNet40 with a single 3090Ti GPU and evaluated using disjoint subsets of ModelNet40. Moreover, we test generalization abilities on the indoor dataset ObjectScanNN and outdoor dataset AirLoc.

The initial source point cloud is transformed by a random transformation via uniform sampling in the Euler angle range $[0, 45°]$ per axis and the translation range $[-0.5, 0.5]$ per axis. The data augmentations we employ follow the approach described in [35]. Specifically, out of the 2048 points, 1024 are randomly and independently subsampled for both the source and target point clouds to generate imperfect correspondences.

We train our model in a two-step process. The model first pretrains on noise-free samples for 16$k$ iterations and then fine-tunes on noisy samples for 16$k$ iterations with a batch size of 32. The noisy points are subjected to jitter with Gaussian noise along the 3D axis by $[-0.05, 0.05]$. We use the AdamW [45] optimizer with weight decay set to $1 \times 10^{-5}$ and clip gradients in the range $[-1, 1]$ during training. We set up 12 iterations of the update operator for all experiments. During iteration, the gradient is backpropagated through the $\Delta T_i$ branch, and zero through the $T_i$ branch.

Without additional training, we test our model on the indoor dataset (ScanObjectNN) and outdoor dataset (AirLoc) for real-world point cloud registration. We do not add additional noise because the datasets are captured from the sensor with real noise.

### 4.3. Baseline Methods

We aim to compare the performance of several registration methods for 3D point clouds. The methods we evaluate include two classical approaches and three learning-based approaches. The classical approaches are the point-to-point iterative closest point (ICP) [21] and fast global registration (FGR) [46]. For learning-based approaches, we evaluate the deep closest Point with transformer (DCP-v2) [47], which is a local feature-based approach that predicts one-shot registration. We also evaluate PointNetLK [13] and ReAgent [18], iterative methods that utilize global PointNet features setting the number of iterations to 10. We retrain PointNetLK and DCP-v2 using published code, as they do not have pretrained models on the ModelNet40 category splits. ReAgent is evaluated by the available pretrained model.

### 4.4. Metrics

Following [18], we report registration accuracy between point cloud pairs by transformation similarity and geometric reprojection error.

Given the ground truth transformation T = [R, t] and the estimated transformation $\hat{T} = [\hat{R}, \hat{t}]$, we employ mean average error (MAE) for measuring the difference between the predicted and actual values of each axis of rotation and translation. The rotation error is calculated using the angular distance between while translation error is calculated using the L2 distance. The MAE of rotation and translation are, respectively, calculated as follows:

$$MAE_x = \frac{1}{3} \sum |x_{gt} - x| \tag{15}$$

where $x$ is either the rotation vector in Euler angles form or the translation vector.

Another metric for measuring the transformation similarity is Isotropic error (ISO). Different from MAE, it describes the error that occurs in the estimation of rotation and translation vectors when the error is equally distributed in all directions. The rotation error $ISO_R$ is computed by:

$$ISO_R = arccos \frac{trace(\hat{R}, R) - 1}{2} \tag{16}$$

and for the translation error $ISO_t$ is computed by:

$$ISO_t = ||\hat{t} - t||_2. \tag{17}$$

In addition to the mentioned metrics, we employ chamfer distance to compare the similarity between two point clouds, which measures the distances between the points in one set to the closest point in the other set:

$$CD(X, Y) = \frac{1}{|X|} \sum_{x \in X} \min_{y \in Y} ||x - y||_2^2. \tag{18}$$

where $x$ and $y$ are the respective points in source $X$ and target $Y$.

The average distance of model points with indistinguishable views (ADI) is proposed in [48], which accounts for true symmetrical transformations by considering the closest point pairs. Given a model under an estimated transformation $X'$ and under the transformation pose $Y$, it is defined as the mean distance between the closet points

$$ADI = \frac{1}{|Y|} \sum_{y \in Y} min_{x' \in X'} ||y - x'||_2 \tag{19}$$

The last metric we employ is the area under the precision–recall curve (AUC) for the average distance of model points with indistinguishable views (ADI) and clops at a precision threshold of 10% of the diameter.

### 4.5. Results

In this section, we will make a detailed comparison of experimental results on three datasets between our model and other methods.

#### 4.5.1. Synthetic Dataset(ModelNet40)

The quantitative results on ModelNet40 are presented in Table 1, where evaluation metrics are reported for two test splits: the first 20 categories of ModelNet40 on the left and the second 20 categories on the right. We achieve an excellent registration result with low MAE, ISO, chamfer distance, and high recall. The results demonstrates our model outperforms the other baseline methods on all performance metrics.

**Table 1. Results on ModelNet40.** We quantitatively compare R-PCR with other baseline methods on held-out point clouds from categories 1–20 (left) and on held-out categories 21–40 (right).

| | Held-Out Models | | | | | | Held-Out Categories | | | | | |
|---|---|---|---|---|---|---|---|---|---|---|---|---|
| | MAE (↓) | | ISO (↓) | | ADI (↑) | $\tilde{CD}$ (↓) | MAE (↓) | | ISO (↓) | | ADI (↑) | $\tilde{CD}$ (↓) |
| | R | t | R | t | AUC | $\times 1e^{-3}$ | R | t | R | t | AUC | $\times 1e^{-3}$ |
| ICP | 3.59 | 0.028 | 7.81 | 0.063 | 90.6 | 3.49 | 3.41 | 0.024 | 7.00 | 0.051 | 90.5 | 3.84 |
| FGR+ | 2.52 | 0.016 | 4.37 | 0.034 | 92.1 | 1.59 | 1.68 | 0.011 | 2.94 | 0.024 | 92.7 | 1.24 |
| DCP-v2 | 3.48 | 0.025 | 7.01 | 0.052 | 85.8 | 2.52 | 4.51 | 0.031 | 8.89 | 0.064 | 82.3 | 3.74 |
| PNLK | 1.64 | 0.012 | 3.33 | 0.026 | 93.0 | 1.03 | 1.61 | 0.013 | 3.22 | 0.028 | 91.6 | 1.51 |
| ReAgent | 1.46 | 0.011 | 2.82 | 0.023 | 94.5 | 0.75 | 1.38 | 0.010 | 2.59 | 0.020 | 93.5 | 0.95 |
| Ours | **0.65** | **0.007** | **2.06** | **0.016** | **96.1** | **0.66** | **0.53** | **0.006** | **1.65** | **0.013** | **96** | **0.72** |

Figure 4 shows the qualitative registration result of our model in comparison with Reagent [18] on ModelNet40. The aligned point cloud transformed by our model appears

more seamless, without any obvious misalignment. Qualitative examples of iterative inference updates are shown in the first line of Figure 5. The transformation estimation network outputs the discrete estimated step sizes and updates the current position of the source in each iteration. During iteration, the step size will adjust dynamically with the position between the source and the target. At the iteration 1, 2, the source adjusts itself in a large step and converges in a small step at last few iterations.

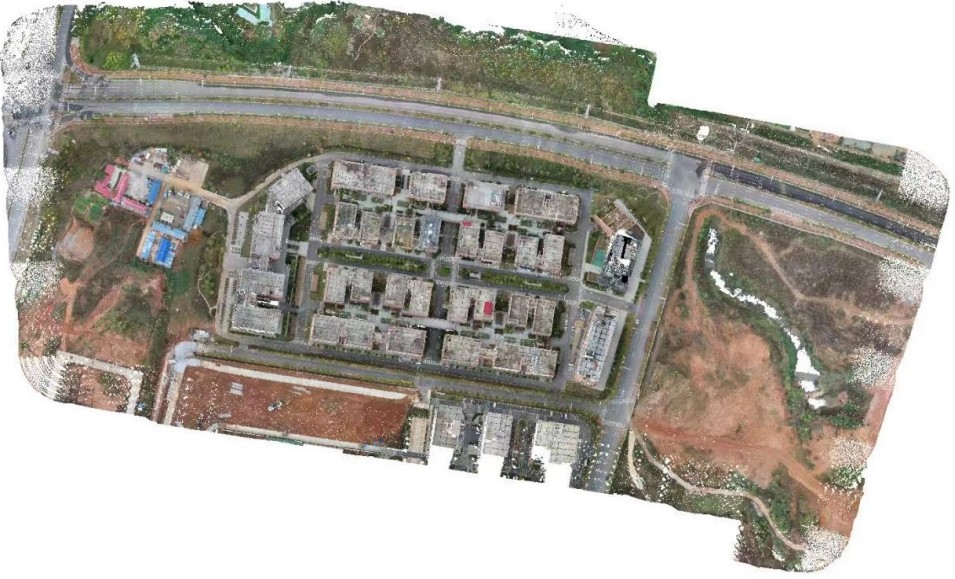

**Figure 3. The map of AirLoc.** The geometric structure of AirLoc is provided by Airborne LiDAR, which allows the quick measurement of distance between LiDAR sensor and the surface of objects it scans, accurately mapping 3D models of outdoor area.

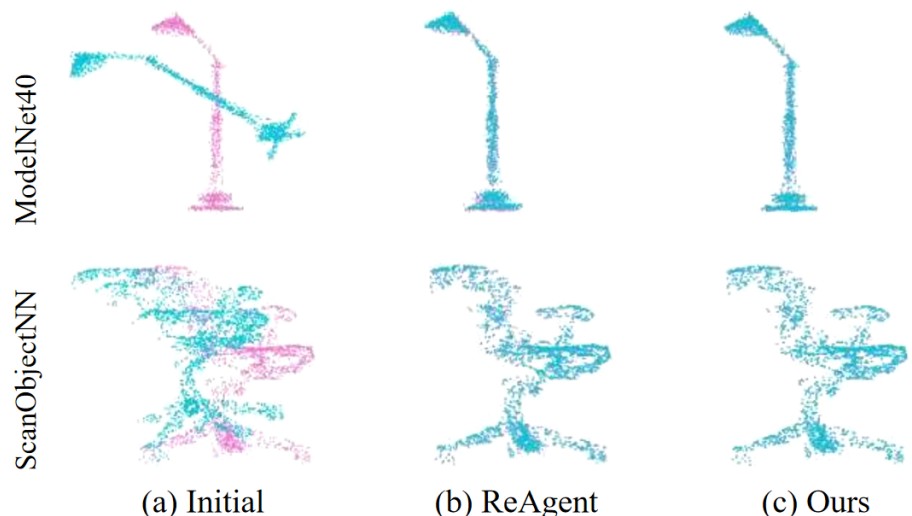

**Figure 4. Qualitative comparisons on ModelNet40 and ScanObjectNN dataset.** Columns show target (magenta) and source (cyan). (**a**) shows the initial state. (**b**,**c**) show the registration results.

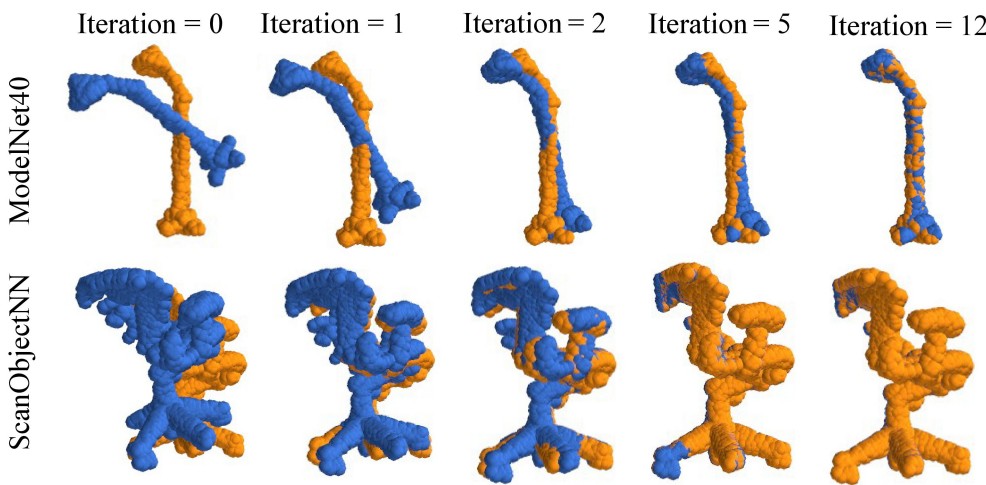

**Figure 5. Qualitative examples on the iterative state of ModelNet40.** The source (**blue**) and the target (**yellow**) aligned over 12 iterations.

### 4.5.2. Indoor Dataset (ScanObjectNN)

Table 2 shows the qualitative results on ScanObjectNN. We compare these four methods, including ICP [21], DCP-v2 [47], PointNetLK [19] and ReAgent [18]. The results demonstrate superior accuracy compared to the other baseline methods across all performance metrics. The second line in Figure 4 shows an example of point cloud registration result on ScanObjectNN. Qualitative examples of iterative inference updates are shown in the second line of Figure 5.

**Table 2. Results on ScanObjectNN.** We quantitatively compare R-PCR with other baseline methods on ScanObjectNN.

| | ScanObjectNN | | | | | |
| | MAE (↓) | | ISO (↓) | | ADI (↑) | CD (↓) |
| | R | t | R | t | AUC | $\times 1e^{-3}$ |
|------|------|-------|-------|-------|------|------|
| ICP | 5.34 | 0.036 | 10.47 | 0.076 | 88.1 | 2.99 |
| DCP-v2 | 7.42 | 0.050 | 14.93 | 0.102 | 72.4 | 4.93 |
| PNLK | 0.90 | 0.010 | 1.74 | 0.020 | 92.5 | 1.09 |
| ReAgent | 0.77 | 0.006 | 1.33 | 0.012 | 95.7 | 0.30 |
| Ours | **0.19** | **0.002** | **0.36** | **0.004** | **97.9** | **0.02** |

### 4.5.3. Outdoor Dataset (AirLoc)

We further verify the validity of our model on the large-scale urban scenes. The results on AirLoc are shown in Table 3. The accuracy and quality of the registration result measured by several metrics all demonstrate the computational efficiency of our model, in the case of large-scale point cloud registration. Figure 6 qualitatively shows the registration results in the laser-scanned map. As shown in Figure 7, we create a convergence curve comparison diagram of each metrics over iterations as test on AirLoc. The curves on the same graph compare the performance of ReAgent and our approach, which show that our approach successfully improves accuracy on the rotation-based metrics and translation metrics.

Additionally, we quantitatively demonstrate the robustness of our approach to different levels of noise-afflicted data. Random noise is sampled from N(0, 0.01), clipped to different margins and applied to the point clouds. We report the improved accuracy across the chamfer distance, as shown in Figure 8. By introducing different levels of noise and comparing the resulting chamfer distances, we gain insight into the robustness of our approach to noise.

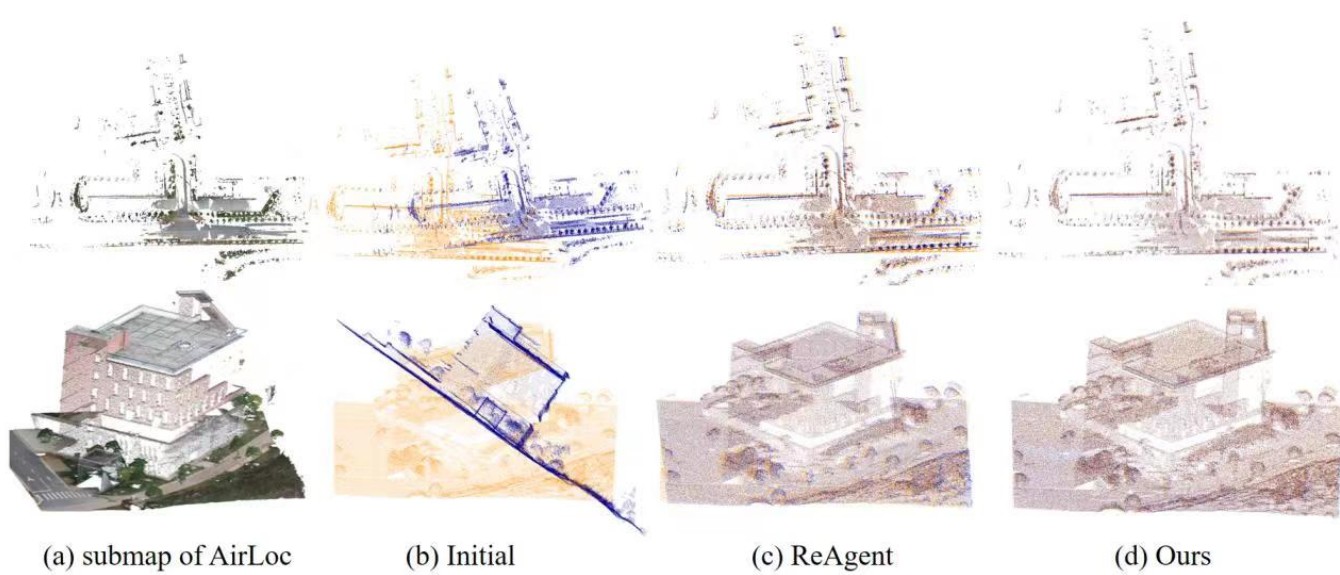

**Figure 6. Qualitative comparisons on AirLoc dataset.** (**a**) shows the submap of laser-scanned submap. (**b**) shows the initial state of point cloud pairs. (**c,d**) shows the registration results.

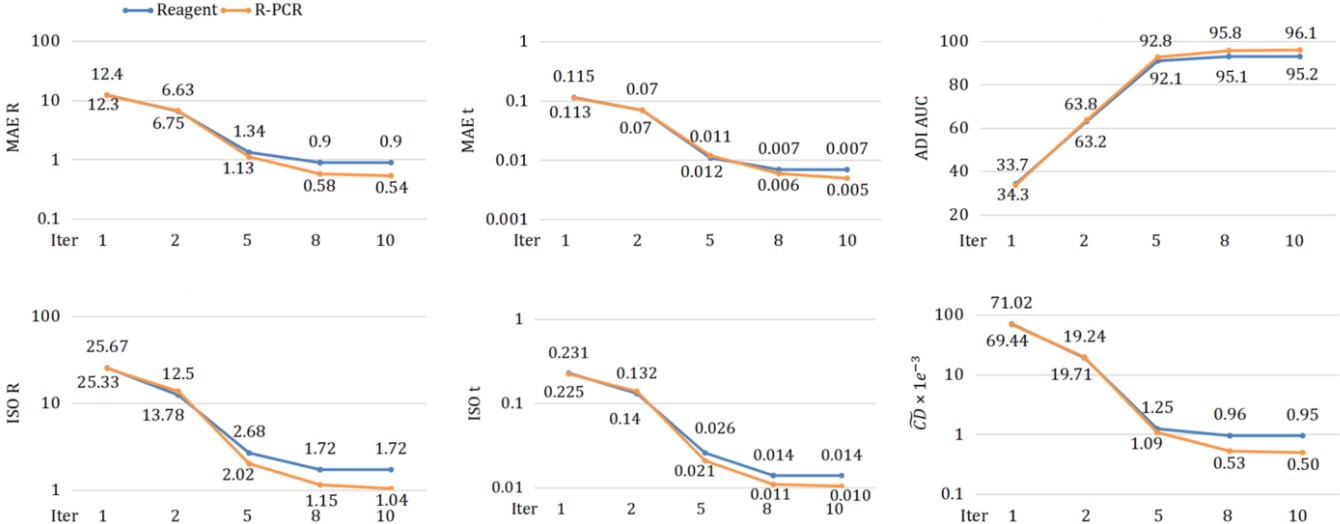

**Figure 7. Convergence curve comparison over iterations on AirLoc dataset.** We plots two curves on the same graph, for ReAgent and R-PCR, with the number of iterations on the x-axis and the value of the metrics on the y-axis. Over iterations, the value of rotation-based metrics, translation metrics and Chamfer distance decrease, with maximal ADI AUC. Compared to ReAgent, R-PCR shows a better convergence performance .

**Table 3. Results on AirLoc.** We quantitatively compare R-PCR with other baseline methods on AirLoc. The registration of our approach aligns the point clouds with minimal error under noisy conditions.

| | AirLoc | | | | | |
| | **MAE (↓)** | | **ISO (↓)** | | **ADI (↑)** | **C̃D (↓)** |
| | **R** | **t** | **R** | **t** | **AUC** | $\times 1e^{-3}$ |
|---|---|---|---|---|---|---|
| ICP | 9.59 | 0.061 | 19.47 | 0.146 | 70.1 | 5.40 |
| DCP-v2 | 9.34 | 0.053 | 18.76 | 0.133 | 73.5 | 4.77 |
| PNLK | 1.43 | 0.012 | 2.38 | 0.020 | 90.3 | 1.29 |
| ReAgent | 1.09 | 0.008 | 1.74 | 0.014 | 93.2 | 0.75 |
| Ours | **0.54** | **0.005** | **1.04** | **0.010** | **96.1** | **0.50** |

### 4.6. Ablation

To further verify the validity of the proposed model, in this section, We perform a set of ablation experiments to show the relative importance of each component. All ablated versions are trained on ModelNet40. The results of the ablations are shown in Table 4. We analyze the effect of each proposed system component in the task of point cloud registration by removing the following module.

### 4.6.1. Recurrent Architecture

We experiment with canceling the GRU cell to make our model become Markovian. Without the GRU cell, all metrics worsen, and the convergence speed is greatly reduced, proving that the gated activation facilitates the convergence of the sequence of transformation.

### 4.6.2. Feature Fusion Module

We try replacing the feature fusion module with a single concatenation operation. The experiment demonstrates that adding a feature fusion module improves the performance of our approach by providing it with a more informative and representative feature for the regression of transformation.

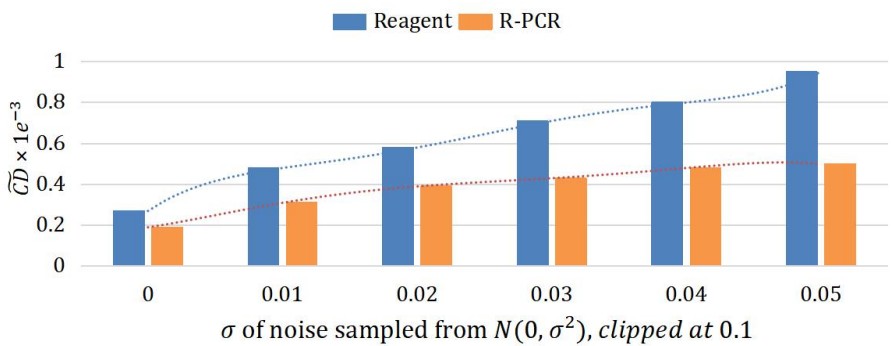

**Figure 8. Results on AirLoc with varying noise.** We introduce different levels of uniform noise to the generated point clouds and evaluate the robustness of the model by comparing the chamfer distances. Our approach shows a smaller increase in chamfer distance as the level of noise is increased.

### 4.6.3. Global Feature Encoding

The input feature vector of the GRU cell is the concatenation of $\psi_i^{fuse}$, hidden state $h_{i-1}$ and high-dimensional transformation feature inherited from step $i-1$. In this experiment, the global fused feature $\psi_i^{fuse}$, concatenated with hidden state, is directly taken as the input of GRU cell without high-dimensional transformation feature and encoding operation. The experiment shows that the global feature encoding module leads to better performance.

### 4.6.4. Sliding Window Size

The sliding window size refers to the receptive field upon local features. We test a range of sizes of sliding windows: 1, 8, 16, and 32. Although R-PCR performs well when the sliding window size is higher than 16, we choose 16 as the final decision owing to the balance between computational complexity and network performance. In addition, we replace the convolution layer with an MLP (multi-layer perception). The result is slightly worse than the convolution counterpart mainly because the parameter redundancy causes an overfitting problem.

### 4.6.5. Iterative Updates

In this experiment, we select a suitable iteration number for updates. We give four maximum thresholds, ranging from 4 to 16. The results show that R-PCR performs best at 12.

**Table 4. Ablation experiments.** We quantitatively validate the effectiveness of different components. See Section 4.6 for detailed descriptions of each of the ablations.

| Method | | Held-Out Models | | | | | Held-Out Categories | | | | | |
|---|---|---|---|---|---|---|---|---|---|---|---|---|
| | | MAE (↓) | | ISO (↓) | | ADI (↑) | ČD (↓) | MAE (↓) | | ISO (↓) | | ADI (↑) | ČD (↓) |
| | | R | t | R | t | AUC | $\times 1e^{-3}$ | R | t | R | t | AUC | $\times 1e^{-3}$ |
| Baseline | | 1.46 | 0.011 | 2.82 | 0.023 | 94.5 | 0.75 | 3.41 | 0.024 | 7.00 | 0.051 | 90.5 | 3.84 |
| recurrent refinement module | | 1.37 | 0.009 | 2.69 | 0.019 | 95.5 | 0.72 | 1.68 | 0.011 | 2.94 | 0.024 | 92.7 | 1.24 |
| Cross concatenation operation | | 1.17 | 0.008 | 2.28 | 0.018 | 95.7 | 0.68 | 0.95 | 0.007 | 1.88 | 0.014 | 95.2 | 0.79 |
| skip connection module | | 0.99 | 0.007 | 2.06 | 0.016 | 96.1 | 0.66 | 0.82 | 0.006 | 1.65 | 0.013 | 96.0 | 0.72 |
| Sliding Window | 1 | 2.11 | 0.015 | 4.30 | 0.031 | 91.2 | 1.28 | 2.27 | 0.015 | 4.43 | 0.032 | 89.5 | 1.69 |
| | 8 | 1.28 | 0.009 | 2.63 | 0.019 | 95.7 | 0.69 | 1.06 | 0.007 | 2.14 | 0.016 | 95.2 | 0.81 |
| | <u>16</u> | 0.99 | 0.007 | 2.01 | 0.016 | 96.1 | 0.66 | 0.75 | 0.006 | 1.52 | 0.012 | 96.3 | 0.70 |
| | 32 | 1.01 | 0.007 | 2.06 | 0.017 | 96.5 | 0.65 | 0.80 | 0.006 | 1.59 | 0.013 | 96.4 | 0.71 |
| | MLP | 1.19 | 0.009 | 2.56 | 0.019 | 95.5 | 0.71 | 1.38 | 0.010 | 2.59 | 0.020 | 93.5 | 0.95 |
| Iterative Updates | 4 | 2.11 | 0.015 | 4.30 | 0.031 | 91.2 | 1.28 | 2.27 | 0.015 | 4.43 | 0.032 | 89.5 | 1.69 |
| | 8 | 1.28 | 0.009 | 2.63 | 0.019 | 95.7 | 0.69 | 1.06 | 0.007 | 2.14 | 0.016 | 95.2 | 0.81 |
| | <u>12</u> | 0.99 | 0.007 | 2.01 | 0.016 | 96.1 | 0.66 | 0.75 | 0.006 | 1.52 | 0.012 | 96.3 | 0.70 |
| | 16 | 1.01 | 0.007 | 2.06 | 0.017 | 96.5 | 0.65 | 0.80 | 0.006 | 1.59 | 0.013 | 96.4 | 0.71 |

## 5. Discussion

In this paper, we propose a novel deep network architecture, R-PCR (recurrent point cloud registration), for large-scale point cloud registration. R-PCR outperforms global-based registration baselines by a large margin on several standard point cloud registration datasets, including synthetic data (ModelNet40) and real data (ScanObjectNN). The effectiveness of R-PCR is demonstrated on large urban data (AirLoc), and it is also shown to be more accurate than the baseline approaches, which is crucial for ensuring stable convergence. We attribute the success to two aspects:

- Efficient feature extraction: R-PCR efficiently fuses independent global features using a PointNet network as an embedding function, which extracts global geometry information by a Siamese structure for source and target point clouds separately. Based on a powerful extractor, our model could learn the feature representations and transformation parameters jointly in an end-to-end fashion.

- Effective global feature fusion: The proposed lightweight cross-concatenation module and large-receptive network merge information between pairwise point clouds, improving the expression ability of global features and introducing possible implicit correspondence, which leads robustness to noise, missing data and could handle a wide range of scenarios.

- High-order Markov decision integration: R-PCR integrates high-order Markov decision into iterative point registration using a recurrent GRU-based update operator. This operator brings high-order state from the previous movement, and the inter-related constraints between substeps model the high-dimensional state and action spaces, making the approach more expressive and better able to model complex registration tasks, particularly noise-afflicted data.

While R-PCR has shown promising results in improving point cloud registration accuracy, there are several areas for further improvement. The R-PCR architecture involves iterative point registration, which could be computationally intensive. Further research can be done to optimize the model's computational efficiency. In addition, although PointNet-based registration is robust to various types of noise and missing data benefits from the shared weights of the PointNet architecture, it may be difficult to achieve the same accuracy as the point clouds have small geometrical consistency. As a result, it may struggle with occluded regions, which could frequently occur in partial overlap registration tasks.

## 6. Conclusions

In this paper, we investigate how to fully leverage the inherent structure and global features to directly estimate the transformation parameters without correspondence. We introduce an end-to-end framework, recurrent point cloud registration, to customize the feature learning for the registration. This method based on high-order Markov significantly has strong stability to align point cloud pairs in noisy conditions, and reaches an optimal result in an energy-saving way.

R-PCR first extracts global geometry information by a PointNet-based Siamese network for source and target point clouds separately. Then, it employs a lightweight cross-concatenation module and large-receptive network merge of information between pairwise point clouds, improving the expression ability of global features and introducing possible implicit correspondence. Furthermore, R-PCR integrates high-order Markov decision into iterative point registration using a recurrent GRU-based update operator. The recurrent units bring high-order state from the previous movement, and the interrelated constraints between substeps model the high-dimensional state and action spaces, making the approach more expressive and better able to model complex registration tasks, particularly noise-afflicted data.

By the comparisons with ModelNet40, ScanObjectNN, and AirLoc, our proposed method outperforms global-based registration baselines by a large margin. The mean average error of rotation and translation of the aligned point cloud pairs is, respectively, reduced by 75% and 66% on the indoor benchmark(ScanObjectNN), and simultaneously by 50% and 37.5% on the outdoor benchmark(AirLoc). The aligned point cloud pairs are visually and geometrically consistent with minimal rotation-based and translation errors under a wide range of scenarios.

**Author Contributions:** Methodology, X.C.; Software, S.Y.; Validation, Y.L.; Formal analysis, M.Z.; Project administration, C.C. All authors have read and agreed to the published version of the manuscript.

**Funding:** This research received no external funding.

**Data Availability Statement:** Our code is available at https://github.com/Choyaa/R_PCR.

**Conflicts of Interest:** The authors declare no conflict of interest.

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
