# Peer review of "R-PCR: Recurrent Point Cloud Registration Using High-Order Markov Decision"

_remotesensing, doi:10.3390/rs15071889_

Round 1

Reviewer 1 Report

 The manuscript clearly presents in a well-structured manne a new method for point cloud registratiom

Author Response

We thank the referee for taking the time to review our manuscript!

Reviewer 2 Report

Remote Sensing: manuscript ID: remotesensing- -2243891 entitled ‘R-PCR: Recurrent Point Cloud Registration Using High-Order Markov’

General comments

The paper concerns a very actual problem accuracy and precision of point clouds registration obtained from different LiDAR sensors and techniques. The Authors rightly point out that the registration process is crucial for establishing a complete scene description and Improving geometric consistency and uncertainty. The novelty and uniqueness of this manuscript is a proposed by Authors a registration scheme which is very suitable for utilizing features extracted from raw point clouds to realize the registrations of scans acquired within the same or across the different LiDAR platforms. The quality and relevance of the datasets appears to be very high. I would like to highlight the potential of this framework and its applicability. This is an interesting topic and application of the methodology, but this paper requires a minor overhaul.

The main shortcoming of this work is its incorrect structure. The authors submitted the manuscript as an article, and this requires a clear separation of the results and discussion sections. Currently, these sections are not present, and the results are contained in several chapters. This needs to be cleaned up so that the authors' findings are in the results section. In the discussion, the authors should demonstrate the superiority of their proposed framework over previously implemented registration methods.

Abstract: The abstract is, in general well written and presents a research idea and results, but it would be better for the reader to add credible numbers to the achievements highlighting the advantages of the current method used by the authors over the other ones.

Conclusions: In fact, this is one conclusion. This is far too little in terms of the authors' findings, but it is also a direct result of the lack of real discussion in this manuscript. With the addition of the discussion, I recommend expanding and adding to the conclusions so that they are better aligned with the authors' findings.

The purpose of the study was formulated as a summary of the results achieved rather than a statement of the research problem undertaken, which the authors intend to solve. The three dotted sentences at the end of the introduction are highlights of achievements rather than the purpose of the paper. This section should be reworded so that the above sentences constitute the 3 specific objectives.

Another issue is the lack of explanation of many of the abbreviations used in the work e.g.: LiDAR, RANSAC, GRU, ICP, RAFT, ADI, MLP. And while some of these are commonly used, the rule of thumb is that before using an abbreviation for the first time, its full name should be provided.

To summarize, in my opinion the study is a "fresh" and reasonable contribution to Remote-Sensing, however, the manuscript needs editorial work. Especially since it does not contain a chapter at all in which the authors discuss their achievements.

Author Response

Thanks for all your feedback and suggestions! We have modified the above suggestions in the latest submitted version. Please download PDF file for detailed response!

Reviewer 3 Report

Dear authors,

The article titled ‘R-PCR: Recurrent Point Cloud Registration Using High-Order Markov Decision’ introduce R-PCR recurrent deep network for point cloud registration employing GRU units with a high-order Markov decision during the iterative refinement process and a cross-concatenation module to fuse point cloud pair features and capture implicit point correspondence information were successfully designed. The experiment results indicate superior performance to non-correspondence methods on indoor and outdoor datasets. Overall, it is an excellent work with scientific soundness. However, I found some points that could be refined enhancing the writing quality of the article as follows:

1.     In line 28, It could be “recent” not “current” because of the works published from 2018 to 2021. I do not think the word is the exact meaning you intended.

2.     In lines 85-86, the mood of the statement ‘claim’ seems not objective.

3.     In line 95, the phrase ‘among others’ could be misused for ‘and so on’.

4.     In lines 124 and 129, the abbreviations such as GRU, RAFT, and so on could be fully represented in a followed bracket when emerging in the context for the first time. It could be desired for beginners.

5.     In lines 166-177, the sentence seems vague.

6.     In line 235, grammar mistake:’…, take an exact…’.

7.     In lines 252-253, equation (14) seems to mistake the comma in the items followed after the first "=".

8.     In line 256, please give the originality of the ModleNet40 dataset, and the corresponding explanation about ObjectScanNN, AirLoc in lines 266 and 267. Readers might be confused when reading here.

9.     In line 261, please add a citation for the AdamW optimizer.

10.  In line 324, adding citations or originalities about DJI M300 and DJI L1 here could be more rigorous in scientific respect.

<The End>

Reviewer

Author Response

(The authors gave the same response as above.)
